# Influence of Geographical Effects in Hedonic Pricing Models for Grass-Fed Cattle in Uruguay

**Paul Harris** [1,*], **Bruno Lanfranco** [2], **Binbin Lu** [3] and **Alexis Comber** [4]

1   Sustainable Agriculture Sciences, Rothamsted Research, North Wyke, Okehampton EX20 2SB, UK
2   Instituto Nacional de Investigación Agropecuaria (INIA), INIA-Las Brujas, Canelones 90200, Uruguay; bruno@inia.org.uy
3   School of Remote Sensing and Information Engineering, Wuhan University, Wuhan 430079, China; binbinlu@whu.edu.cn
4   School of Geography, University of Leeds, Leeds LS2 9JT, UK; a.comber@leeds.ac.uk
*   Correspondence: paul.harris@rothamsted.ac.uk

**Abstract:** A series of non-spatial and spatial hedonic models of feeding and replacement cattle prices at video auctions in Uruguay (2002 to 2009) were specified with predictors measuring marketing conditions (e.g., steer price), cattle characteristics (e.g., breed) and agro-ecological factors (e.g., soil productivity, water characteristics, pasture condition, season). Results indicated that cattle prices produced under extensive production systems were influenced by all of predictor categories, confirming that found previously. Although many of the agro-ecological predictors were inherently spatial in nature, the incorporation of spatial effects into the estimation of the hedonic model itself, through either a spatially-autocorrelated error term or allowing the regression coefficients to vary spatially and at different scales, was able to provide greater insight into the cattle price process. Through the latter extension, using a multiscale geographically weighted regression, which was the most informative and most accurate model, relationships between cattle price and predictors operated at a mixture of global, regional, local and highly local spatial scales. This result is considered a key advance, where uncovering, interpreting, and utilizing such rich spatial information can help improve the geographical provenance of Uruguayan beef and is critically important for maintaining Uruguay's status as a key exporter of beef with respect to the health and safety benefits of natural, open-sky, grass-fed production systems.

**Keywords:** beef cattle prices; spatial regression; multiscale; provenance; MGWR

## 1. Introduction

Cattle herds have heterogenous characteristics and distinct qualitative differences. Some of these can be measured through market price premiums and discounts and associated attributes, allowing herd properties to be differentiated. Cattle traders assign financial values to different combinations of herd characteristics as well as herd provenance. The latter relates to the specific region of production, and regional agro-ecological characteristics. These may result in price premiums or discounts because they provide summary indicators of the permanent conditions under which livestock are produced affecting both the cattle and the final meat product [1]. Other, temporary agro-ecological conditions (such as precipitation) may influence short-term supply and demand and price, due to producers' contraction or expansion decisions. Thus, cattle market prices reflect both herd- and region-specific properties, as well as individual trader objectives or preferences, and cattle condition and appearance at the time of auction [2,3].

In a seminal paper, Rosen [4] formulated the theory of hedonic prices as an economic problem of equilibrium, where a complete set of implicit prices guide the decisions of buyers and sellers in a *k*-dimensional space of traits. In this context, Rosen defined hedonic prices as the implicit prices of traits, which are revealed by the economic agents through the observed prices of differentiated products, and the specific quantities of the traits associated to them. Following this idea, the price at which live cattle is bought and sold in the market can be expressed as a function that relates the market price with the quality embedded in a lot of cattle, measured in terms of its quality traits [4–8]. Subsequently, many authors have applied alternative and improved versions of hedonic models with differentiated market products [7,9–11].

Through a hedonic model, Lanfranco et al. [2] evaluated the impact of livestock characteristics, market strategies and conditions on live cattle prices in Uruguay. For some sales (lots), they found that body weight, sex, predominant breed, the presence of certain crossbreeds and animal size uniformity to be the characteristics most important to buyers. While these attributes can be objectively measured, subjective visual assessments of livestock quality also influence livestock prices and perceived value. Additionally, the order in which cattle are auctioned and the number of animals in a particular lot influence market price. In the case of video auctions, buyers view the videos remotely and rely on visual assessments of the cattle by inspectors. The inspectors certify the lots and make recommendations which are highly valued by video auction buyers. Their decision-making is also influenced by market conditions, exchange rates, and long-term price expectations within the sector [2].

Overall, the prices at which cattle are bought and sold are influenced by cattle attributes, overall market conditions, and marketing strategies [6,12–15]. However, significant regional and seasonal factors also affect cattle prices [2], which are recognized by cattle producers and other actors in national beef supply chains. There can be significant differences between the prices paid for cattle, *ceteris paribus*, because of their geographic origins [2]. This "provenance effect" encapsulates different variables that influence nutrition and management and in turn, cattle development and condition, and therefore affect livestock prices. Soil, forage types, typical practice, and production conditions are driven by regional provenance due to the heterogeneity of the agro-ecological characteristics of land supporting grass-fed cattle production. The result is that cattle prices, the income derived from cattle sales, and the land value of areas used for cattle production (with implications for understanding of the ranch real estate market) will vary geographically. For these reasons, Lanfranco and Castaño [3] extended the hedonic cattle price model of Lanfranco et al. [2] to include both permanent and temporary agro-ecological factors. They found that dominant agro-ecological conditions (soil characteristics, water availability, and average seasonal climatic conditions) could explain the geographical patterns of live cattle prices. In addition, weather variability and especially extreme climatic events were found to have important short-term impacts on cattle markets—as pasture conditions improved or precipitation increased (e.g., both soil moisture and surface runoff), livestock prices tended to fall, *ceteris paribus*. In this way, Lanfranco and Castaño [3] increased the understanding of how geographical factors affect output price and the potential asset-value impacts of agro-ecological variability in Uruguayan grass-fed beef production.

However, although the study of Lanfranco and Castaño [3] described the geographical patterns in cattle price, informed through agro-ecological factors, the parameters of the hedonic model itself were estimated without due consideration for spatial effects. As the cattle production process is inherently spatial, such naive model estimations can result in both poor prediction and inference. In this respect, the objectives of this current study were to spatially-adapt the (extended) non-spatial model of Lanfranco and Castaño [3]. Two distinct spatial adaptations were considered: (i) a regression model that accounts for spatial autocorrelation effects in the error term [16]; and (ii) a regression model that accounts for spatial heterogeneity effects in the response to predictor variable relationships [17–19]. The use of spatially-explicit hedonic models is common in the house and land price literature [20–29], from which this current study transfers concepts and ideas. Thus, three hedonic regressions were specified, a non-spatial

model (mimicking previous work) and two spatial adaptations, to provide a novel and spatially-informed comparison of hedonic price models for the Uruguayan cattle markets.

## 2. Methods

### 2.1. Study Data

#### 2.1.1. Spatial Unit and Cattle Price

As in Lanfranco and Castaño [3], cattle were geographically identified by their Uruguayan police precinct of origin (an administrative division at the sub-department level), where precinct boundaries were assumed to approximately define homogeneous agro-ecological conditions. Cattle price data were compiled from the three largest video auctions operating in Uruguay: Pantalla Uruguay, Plaza Rural and Lote 21. Prices are in US dollars and are expressed per kilogram of live weight (US$/kg LW). The focus of the analysis was on market transactions for cattle lots sold by weight, where lots commonly consisted of all cows, all heifers, or all steers destined for either herd replacement or fattening. The full database consisted of $N = 14,476$ individual registered lots from 91 video auctions between June 2002 and June 2009. Pantalla Uruguay provided data for 23 video auctions from 11 June 2002 to 27 December 2004; Plaza Rural, data for 43 auctions from 2 July 2003 to 19 December 2007; Lote 21, data for 25 auctions from 25 October 2005 to 9 June 2009. Details on adjusting cattle price to account for prevalent market conditions are given in Lanfranco and Castaño [3]. Although the data have a strong temporal component, this and previous studies, did not seek to model this aspect directly, but only indirectly through an "averaged" seasonal effect on cattle price.

#### 2.1.2. Market and Cattle Characteristics

The predictor variables for market conditions, marketing strategies and livestock characteristics (Table 1), were the same as that used by Lanfranco et al. [2]; Lanfranco and Castaño [3]. At the moment of each sale, the market conditions were represented by average weekly slaughter prices of fat steers (obtained from Uruguay's National Meat Institute) and the exchange rate between the US dollar and the Uruguayan peso. Prices are again expressed in US$/kg LW. Variables that related to cattle attributes for each lot and the auction marketing strategies were taken from the respective auction catalogs. For descriptive variables, this was done indirectly, using proxies, while, for quantitative variables, it was done directly.

#### 2.1.3. Agro-Ecological Conditions

Following Lanfranco and Castaño [3], this study similarly distinguished between permanent and temporary agro-ecological conditions affecting live cattle markets (Table 2). Two predictors represent permanent agro-ecological conditions, the CONEAT index (a soil productivity index devised by the Comisión Nacional de Estudio Agronómico de la Tierra) [30] and soil water-holding capacity (WHC, expressed in millimeters of water in the soil profile). CONEAT and WHC for each police precinct were computed from area weighted soil compositions. The temporary agro-ecological conditions were described by the season of sale (summer (T1), fall (T2), winter (T3) and spring (T4)), and three further predictors. First, the normalized difference vegetation index (NDVI) represented available pasture forage (pasture condition). NDVI provides an estimate of vegetative cover obtained through satellite remote sensing, where average NDVI values for the month prior to each lot's sale were used. Water availability, both for cattle consumption and pasture evapotranspiration, was accounted for by PAW (percentage of available water in the soil profile relative to field capacity) and SWR (surface water runoff). Values of WHC and SWR were estimated for the quarter immediately preceding each auction.

**Table 1.** Market and cattle characteristics used as predictor variables.

| Predictor Variables | Name | Type | Comments |
|---|---|---|---|
| **Beef and General Market Conditions** | | | |
| Steer price (US$/kg LW) | PSTEER | Numeric | Slaughter price of steers at sale |
| Exchange rate (UY$/US$) | EXRT | Numeric | General market conditions at sale |
| **Auction Marketing Strategy** | | | |
| Order of entry (#) | ORDER | Numeric | Order in which the lot was auctioned ** |
| Lot size (#) | LOTSZ | Numeric | Size of the cattle lot ** |
| Recommended lot (Yes/No) | RECOM | Binary | Lot explicitly recommended by inspector |
| **Cattle Attributes** | | | |
| Males (Yes/No) | MALE | Binary | Lot 100% composed by male calves or steers |
| Live weight (kg) | KLW | Numeric | Average weight of the animals in the lot ** |
| Class (scored 3 to 10) | CLASS | Ordinal Set | Class of animals (regular to excellent) |
| Condition (scored 3 to 10) | COND | Ordinal Set | Condition of animals (regular to excellent) |
| Age uniformity (Yes/No) | AGEU | Binary | Uniformity of cattle lot according to age |
| Shape uniformity (Yes/No) | UNIF | Binary | Uniformity according to size, frame, etc. |
| Improved nutrition (Yes/No) | INUT | Binary | Cattle lot receiving improved nutrition level |
| Tick area (Yes-high/Yes-low/No) | TKAR | Ordinal Set | Lot from tick-infested area. 0—No (no risk); 1—Yes (no ticks—low risk); 2 Yes (ticks—high risk) |
| Mio-Mio (Yes/No) | BCAR | Binary | Lot from area infested with *Bacharis coridifolia* |
| Predominant breed (Yes/No) | BD1–BD6 | Binary Set | 1—Hereford *; 2—Angus; 3—Other British; 4—Continental; 5—Dairy; 6—Zebu |
| Crossbreeds (Yes/No) | CZ1–CZ3 | Binary Set | Hereford/Angus (CZ1), British/Continent. (CZ2), Dairy/Zebu (CZ3) |
| **Interactions between predictors** | | | |
| Lot size × Weight (kg) | LXW | Numeric | Interaction between weight and lot size |
| Condition × Weight (kg) | CXW | Numeric | Interaction between condition and lot size |

*—Omitted composite binary variable; **—Variable included in both linear and quadratic form; #—Unitless.

**Table 2.** Agro-ecological conditions used as predictor variables.

| Predictor Variables | Name | Type | Comments |
|---|---|---|---|
| **Permanent** | | | |
| Soil productivity (#) | CONEAT | Index | Soil productivity (CONEAT index) **. It measures the productivity (in terms of meat) of any piece of rural land according to the proportion of soils (composition, fertility, slope, physical structure). |
| Water hold. capacity (mm) | WHC | Numeric | Capacity of holding water in soil profile ** |
| **Temporary** | | | |
| Season of sale (Yes/No) | T1–T4 | Binary Set | 1—Summer; 2—Fall; 3—Winter; 4—Spring * |
| Pastures condition (#) | NDVI | Index | Normalized Difference Vegetation Index **. It takes values between 0 and 100, so is compositional in form. |
| Surface water runoff (mm) | SWR | Numeric | Water runoff (not penetrating in soil) ** |
| Available water (%) | PAW | Percentage | Water already available in soil profile ** |
| **Interactions between temporary predictors only** | | | |
| SWR × PAW | SXP | Numeric | Interaction of SWR and PAW |
| NDVI × Season of the year | NXT1-T3 | Numeric | Interaction of pasture condition and season |
| SWR × Season of the year | SXT1-T3 | Numeric | Interaction of water runoff and season |
| PAW × Season of the year | PXT1-T3 | Numeric | Interaction of water in soil and season |

*—Omitted composite binary variable; **—Variable included in both linear and quadratic form; #—Unitless.

## 2.2. Statistical Models

Details of the study regressions are presented in the Supplementary Materials (SM, Section S.1), where the estimation of a linear regression (LR), a linear mixed model (LMM) [31] and a multiscale geographically weighted regression (MGWR) [27,32–34] are described. Models are estimated via ordinary least squares (OLS), restricted maximum likelihood (REML), and an iterative weighted least squares back-fitting procedure for the LR, LMM and MGWR, respectively. The LMM and the MGWR extend the hedonic LR model of Lanfranco and Castaño [3] consisting of 50 predictor variables (Tables 1 and 2) that include quadratic as well as interaction terms. The LR is a non-spatial fixed coefficient model, the LMM is also a fixed coefficient model but accounts for residual spatial autocorrelation from the LR, while MGWR is a spatially varying coefficient (SVC) model where each set of SVCs can operate at their own spatial scale. Thus, MGWR captures change in response to predictor variable relationships across space, while the LR and the LMM do not. Spatial structure (autocorrelation) is identified through the residual variogram in the LMM, while spatial structure (heterogeneity) is identified via a kernel weighting function in MGWR where a different kernel bandwidth is estimated for each data relationship. Model outputs are reported in terms of model fit ($R^2$ and Akaike Information Criterion (AIC) values) and coefficient significance, where LR and LMM estimate a single coefficient per relationship, while MGWR estimates multiple coefficients per relationship that can be mapped. In summary, the following regression form was specified for LR, LMM and MGWR:

$$PRICE = f(Beef\ and\ Market\ Conditions + Auction\ Strategies + Cattle\ Attributes + Agroecological\ Conditions) \quad (1)$$

## 2.3. Data Preparation

Parameters for the LR, LMM, and MGWR were estimated using data for lots that were actually auctioned (i.e., observations with cattle price equal to zero were removed); the revised database was comprised of $N = 12{,}523$ observations (down from $N = 14{,}476$), which was further reduced to remove observations with missing values (to $N = 12{,}382$ observations). This entailed that 274 spatial units (police precinct) were reduced to 229 spatial units, where each spatial unit contained from a minimum of a single observation to a maximum of 285 observations. This unbalanced spatial information presented a preferential sampling issue and associated bias in model estimation [35]. A pragmatic approach to this problem was adopted where a stratified random sample was taken (with the spatial unit being the stratum) ensuring at most 15 observations per spatial unit, yielding a revised study data set of $N = 2845$ observations (see SM, Section S.3). This level of data decimation also eased the computational burden on fitting the LMM and the MGWR. The resultant mean and standard deviation of cattle price per spatial unit are mapped in Figure 1, where missing data commonly occurred in urban areas, as would be expected given the study subject (for example, Montevideo in the South). To avoid problems of singularity in the data matrix due to the presence of a regression constant (exact collinearity), one option was discarded for the composite binary variables (breed and season) and their possible interactions. For the simple binary variables, the default option has the variable set to zero. A small random jitter (error) to the season (fall and winter only) and the tick area binary variables was also applied to avoid problems of localized singularity with the MGWR fit [36]. All distance calculations for the LMM and MGWR were based on the police precinct centroids.

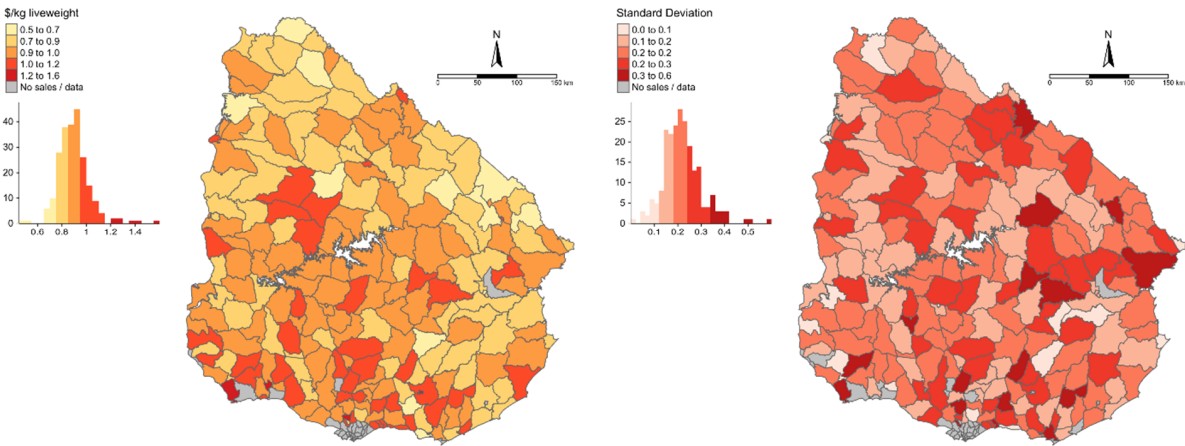

**Figure 1.** Mean and standard deviation of cattle price using the decimated data set ($N = 2845$), each given with *k*-means breaks.

## 3. Results

Study results with respect to stationary (or global) cattle price relationships are reported in detail in SM, where only the salient results need reporting in this main text section. This is because, to some extent, such relationships have been reported elsewhere [2,3]. This approach enables the main text to focus on the investigation of identified nonstationary (or local) cattle price relationships and the spatial scales they operate at.

### 3.1. Estimation of the LR Model

Tests of joint significance of the LR coefficients associated with the variables of interest resulted in rejection of the null hypothesis ($H_0$: $\beta_1 = \beta_2 = ... = \beta_{51} = 0$), indicating that the LR model had good explanatory power. The $R^2$ statistic was strong at 0.85, and the model's AIC statistic was calculated at –5144. Summaries for the LR are presented in Tables S1 to S2 in SM (Section S.2). With respect to the hypothesis $H_0$: $\beta_k = 0$ and Ha: $\beta_k \neq 0$, for $k = 1, ..., 51$, the results indicated that 29 of the 51 estimated coefficients for the LR were significantly different from zero, at least at the 5% ($\alpha = 0.05$) level. Here, 14 significant relationships reflected market and cattle characteristics, while 14 significant relationships reflected agro-ecological conditions.

### 3.2. Estimation of the LMM

The LMM $R^2$ value was little different to its LR counterpart with an $R^2 = 0.85$. The AIC statistic was however "strongly" lowered in comparison to the LR fit, with an index of –5168 (i.e., yielding an improvement of 24 units). The variogram parameters for the LMM (see SM, Section S.2) were estimated at 0.926 for the nugget effect (i.e., $c_0 / (c_0 + c_1)$) and 1213.2 m for the correlation range (*a*). A high nugget effect (the maximum = 1) combined with a relatively short correlation range indicted that only weak spatially-autocorrelated errors were observed. This ensured the LMM was likely to be similar in nature to its LR counterpart, where, for the latter, random errors are assumed. Summaries for the LMM are presented in Tables S3 and S4 in the SM (Section S.2) Here, 27 of the 51 estimated coefficients were significantly different from zero (at 5% level), where now 12 of the 23 coefficients for the agro-ecological predictors, were significant. It is important to review how an LMM relates to its corresponding LR in terms of significant predictors, where significant predictors in the LR model often become insignificant in the LMM, if they themselves exhibit strong spatial trends [37]. In this respect, two significant predictors (at 5% level)

in the LR model—SWR (surface water runoff) and PAW (available water, in quadratic form only, PAW2) each became insignificant in the LMM.

### 3.3. Estimation of the MGWR Model

For MGWR, the model was first run with mean centered predictor variables [38] to estimate the bandwidths, then, secondly, the model was re-run with the bandwidths from the first run fixed (i.e., not re-estimated) but now with the raw predictors. The first run with centered data ensured that bandwidth estimation was not compromised by a set of predictors measured on different scales. Such data pre-processing also reduces the computational burden in the MGWR calibration. The second run with raw data allows the MGWR parameters to be directly comparable to the LR and LMM parameters (also calibrated with the raw data) and maintains the properties of a hedonic price model. For the resultant MGWR fit, the $R^2$ = 0.89; thus, MGWR provided the most accurate (*in-sample*) hedonic model fit. The AIC statistic for MGWR was estimated at −5426, yielding an improvement of 282 and 258 units over LR and LMM, respectively. The MGWR model took a considerable amount of computation to fit relative to the LMM and the LR fits, where the MGWR took days, the LMM took hours, while the LR took seconds.

### 3.3.1. Bandwidths and Identified Nonstationary Relationships

In total, 15 out of a possible 51 relationships were considered nonstationary in MGWR, where those considered nonstationary had estimated kernel bandwidths much lower than the maximum distance between sample locations of 564 km. Bandwidths estimated close to, or at this maximum distance were considered global (i.e., represent stationary relationships) and, in a pragmatic sense, should be fixed as such. Identified nonstationary relationships were for cattle price with the intercept (C), PSTEER (steer price), MALE (male), KLW (live weight), UNIF (shape uniformity), INUT (improved nutrition), BD2 (Angus), BD4 (Continental), BD6 (Zebu), CZ2 (British Continental cross), T1 (summer), T3 (winter), NDVI (pasture condition), the interaction between SWR (surface water runoff) and T2 (fall) (SXT2), and the interaction between SWR and T3 (SXT3). The relationship between cattle price and PSTEER operated at a very local scale with a bandwidth of only 6 km, while cattle price's relationship to the intercept, MALE, BD2, T3 and SXT2 operated at relatively small spatial scales with respective bandwidths of 54, 94, 122, 127 and 109 km. Conversely, the relationships between cattle price and KLW, UNIF, INUT, BD4, BD6, CZ2, T1, NDVI and SXT3, each operated at relatively large spatial scales with respective bandwidths of 506, 485, 374, 320, 486, 296, 295, 262 and 242 km. Observe that PSTEER, MALE, KLW, UNIF, INUT, BD2, BD4, BD6 and CZ2 represent market and cattle characteristics, while T1, T3, NDVI, SXT2 and SXT3 represent temporary agro-ecological conditions.

### 3.3.2. Stationary Relationships and Their Significance

Estimated stationary regression coefficients and their significance from zero at the 5% level are presented in Table S5 in SM (Section S.2). Given the respective bandwidths each tend to the maximum, coefficient and the *p*-value distributions deviate little from their respective medians or means, and the former is reported. Here, 17 out of the 36 stationary coefficients for MGWR were significantly different to zero. The same 17 coefficients were all also significantly different to zero for the LR and LMM fits, which provides assurance and consistency across all three models. Thus, exchange rate (EXRT), order of entry (both ORDER and ORDER2), lot size (LOTSZ), recommended lot (RECOM), live weight quadratic (KLW2), class (CLASS), Dairy breed (BD5), Hereford Angus cross (CZ1), water holding capacity (WHC), NDVI quadratic (NDVI2), surface water runoff quadratic (SWR2), available water (PAW), the interaction between SWR and PAW (SXP), the interaction between NDVI and summer and NDVI and fall (NXT1 and

NXT2, respectively), and the interaction between PAW and summer (PXT1) each exert a global or national influence on cattle price where the respective relationship is invariant across space.

### 3.3.3. Nonstationary Relationships and Their Significance

For the nonstationary coefficients of MGWR (C, PSTEER, MALE, KLW, UNIF, INUT, BD2, BD4, BD6, CZ2, T1, T3, NDVI, SXT2, and SXT3), only C, PSTEER, MALE, KLW, BD2, NDVI, and SXT2 were significantly different to zero for the LR and LMM fits. The spatial distributions of the coefficients that were estimated as nonstationary are mapped in Figure 2 (market and cattle predictors only) and Figure 3 (temporary agro-ecological predictors only plus the intercept), where their significance from zero (5% level) is highlighted.

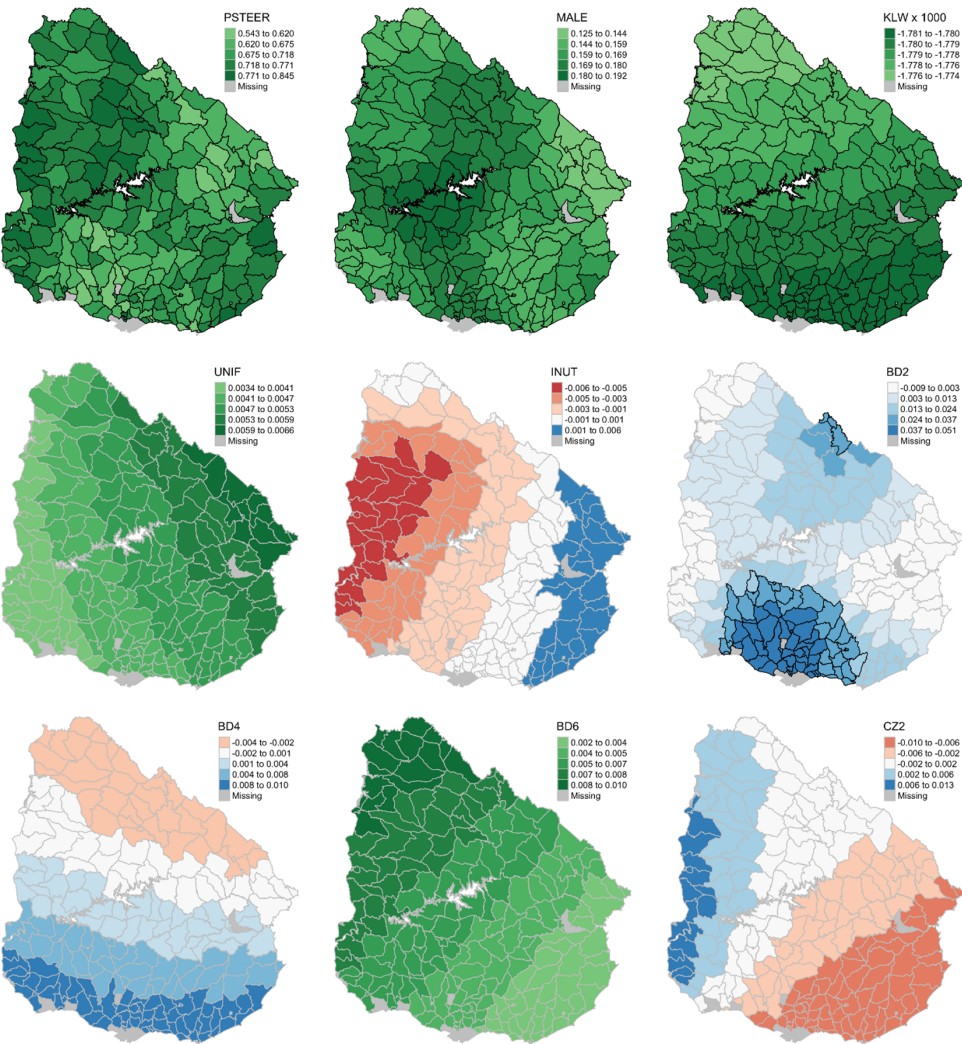

**Figure 2.** SVCs for market and cattle characteristics: PSTEER (steer price), MALE (male), KLW (live weight), UNIF (shape uniformity), INUT (improved nutrition), BD2 (Angus breed), BD4 (Continental breed), BD6 (Zebu breed), and CZ2 (British Continental cross). The SVCs are shown with their significance from zero by highlighting the border of their areal unit (otherwise not significant at 5% level) and have all been corrected for multiple comparisons.

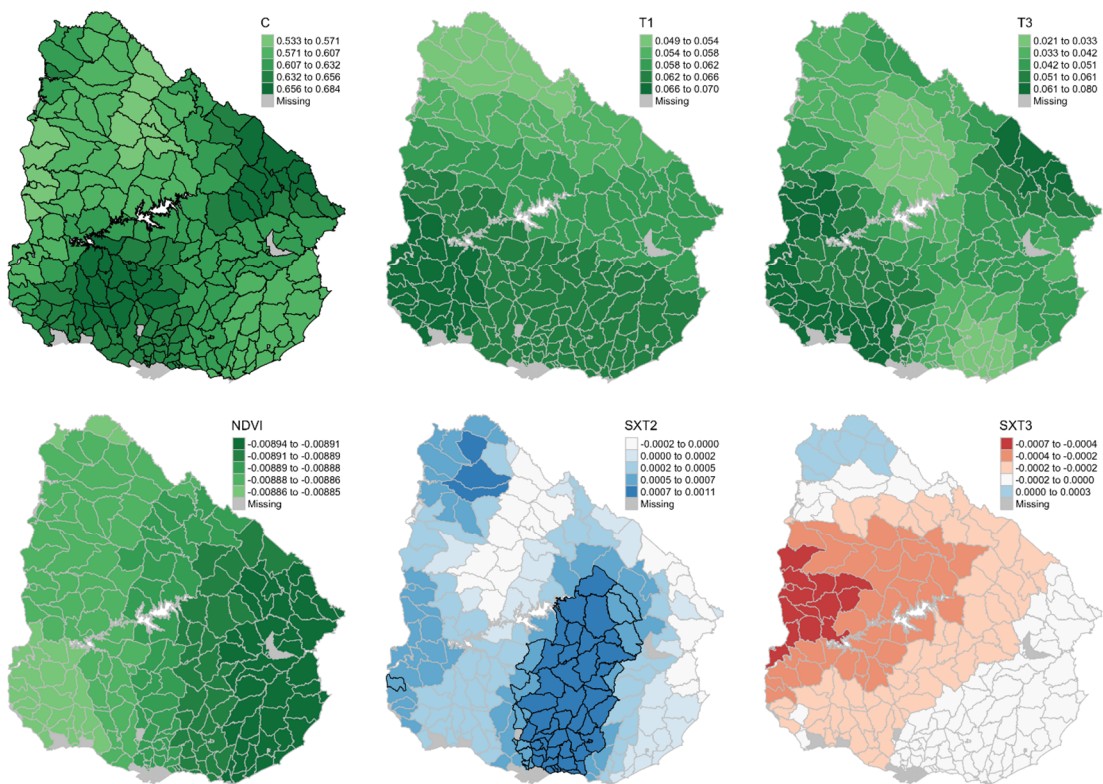

**Figure 3.** SVCs for temporary agro-ecological conditions: The intercept (C), T1 (summer), T3 (winter), NDVI (pasture condition), the interaction between SWR (surface water runoff) and T2 (fall) (SXT2), and the interaction between SWR and T3 (summer) (SXT3). The SVCs are shown with their significance from zero by highlighting the border of their areal unit (otherwise not significant at 5% level) and have all been corrected for multiple comparisons.

Discussing each nonstationary relationship in turn, the coefficient for PSTEER was estimated as stationary, positive, and significant for LR/LMM, while, for MGWR, it was estimated as nonstationary, positive, and significant everywhere. The behavior of the SVCs for PSTEER was erratic (reflecting its very localized bandwidth of 6 km), where there were broadly lower contributions towards cattle price in central regions moving Northeast to the central South. The MALE coefficient was estimated as stationary, positive, and significant for LR/LMM, while for MGWR, it was estimated as nonstationary and significantly positive everywhere, with weaker influences on cattle price in Eastern, Northwestern, and Southwestern regions. The KLW coefficient was estimated as stationary, negative, and significant for LR/LMM, while, for MGWR, it was estimated as nonstationary, negative, and significant everywhere, with a weakening influence on cattle price moving in a clear trend from South to North.

The UNIF coefficient was estimated as stationary, negative, and insignificant in the LR, stationary, positive, and insignificant in the LMM, while, for MGWR, it was estimated as nonstationary, positive, and insignificant everywhere (with a weakening influence on cattle price moving East to West). The INUT coefficient was estimated as stationary, negative, and insignificant for LR/LMM, while, for MGWR, it was estimated as nonstationary, both negative and positive, and insignificant everywhere (with a weakening influence moving East to West). The BD2 coefficient was estimated as stationary, positive and significant in the LR/LMM, while, for MGWR, it was estimated as nonstationary and almost exclusively positive. The BD2 predictor provided the first instance where the SVCs were only sometimes significant, where the

highest contributions towards cattle prices were in central Southern regions surrounding Montevideo, coinciding with coefficient significance.

The BD4 coefficient was estimated as stationary, positive, and insignificant for LR/LMM, while, for MGWR, the coefficients were estimated as nonstationary, positive (in the South), and negative (in the North), but insignificant everywhere. The BD6 coefficient was estimated as stationary, positive, and insignificant for LR/LMM, while, for MGWR, the coefficients were estimated as nonstationary, positive, and insignificant everywhere (moving in a Southeast to Northwest trend). The CZ2 coefficient was estimated as stationary, positive, and insignificant for LR/LMM, while, for MGWR, it was estimated as nonstationary, both negative and positive, and insignificant everywhere (moving in an East to West trend).

For the temporary agro-ecological relationships to cattle price, the coefficients for T1 and T3 were both estimated as stationary, negative, and insignificant for LR/LMM, while, for MGWR, they were estimated as nonstationary, this time both positive, but again insignificant everywhere (coefficients for T1 displayed a North to South trend while coefficients for T3 were higher in the Southwest and the Northeast). For the NDVI, the coefficient was estimated as stationary, negative, and significant for LR/LMM, while, for MGWR, it was estimated as nonstationary, negative, but now insignificant everywhere (significance aside, NDVI had the greatest influence on cattle price in Eastern regions of Uruguay). For the interaction term SXT2, the coefficient was estimated as stationary, positive, and significant for LR/LMM, while, for MGWR, it was estimated as nonstationary, both positive and negative, and both significant (for positive relationship only) and insignificant, with its greatest influence on cattle price centrally and to the South. The SXT3 coefficient was estimated as stationary, negative, and insignificant for LR/LMM, while, for MGWR, it was estimated as nonstationary, both positive and negative, but always insignificant (where its greatest influence on cattle price occurred in a central Western region). The behavior of the intercept (C) is discussed in Section 4.

### 3.4. Summary

From model $R^2$ and AIC values (Table 3), MGWR was the best fitting model, highlighting clear value in considering scale-dependent, nonstationary relationships. For local fit performance, maps of the mean and standard deviation of the residuals from the model fits are given in Figure 4, where MGWR clearly provided the lowest means and in general, lower standard deviations. From all results (fit and coefficient significance), the LMM provided spatial clarity on (assumed) stationary relationships in comparison to LR, noting that SWR and PAW2 were significant predictors in the LR but insignificant in the LMM.

**Table 3.** Summary of study results.

| Model | Spatial Effects? | $R^2$ | AIC | Intercept Behavior |
|---|---|---|---|---|
| LR | No | 0.85 | −5144 | Stationary and significant * |
| LMM | Yes | 0.85 | −5168 | Stationary and significant * |
| MGWR | Yes | 0.89 | −5426 | Nonstationary and significant at all locations * |

* Significant at the 5% level or lower.

The status of the relationship between cattle price and a given predictor depended on the model used (Table 4), where MGWR provided the greatest change in relationship status. Significant relationships, that remained consistently stationary with cattle price across all three models, were EXRT, ORDER, ORDER2, LOTSZ, RECOM, KLW2, CLASS, BD5, and CZ1, all for market and cattle characteristics; WHC for permanent agro-ecological factors; and NDVI2, SWR2, PAW, SXP, NXT1, NXT2, and PXT1 for temporary agro-ecological factors. Significant relationships estimated as nonstationary via MGWR were for the intercept (C), PSTEER, MALE, KLW, BD2, and SXT (thus no relationships were estimated as such for the permanent agro-ecological factors). NDVI is the only predictor that was estimated as significant in

LR/LMM but insignificant (and nonstationary) in MGWR. The behavior of the intercept across the three models is summarized in Table 3, where it was always significant.

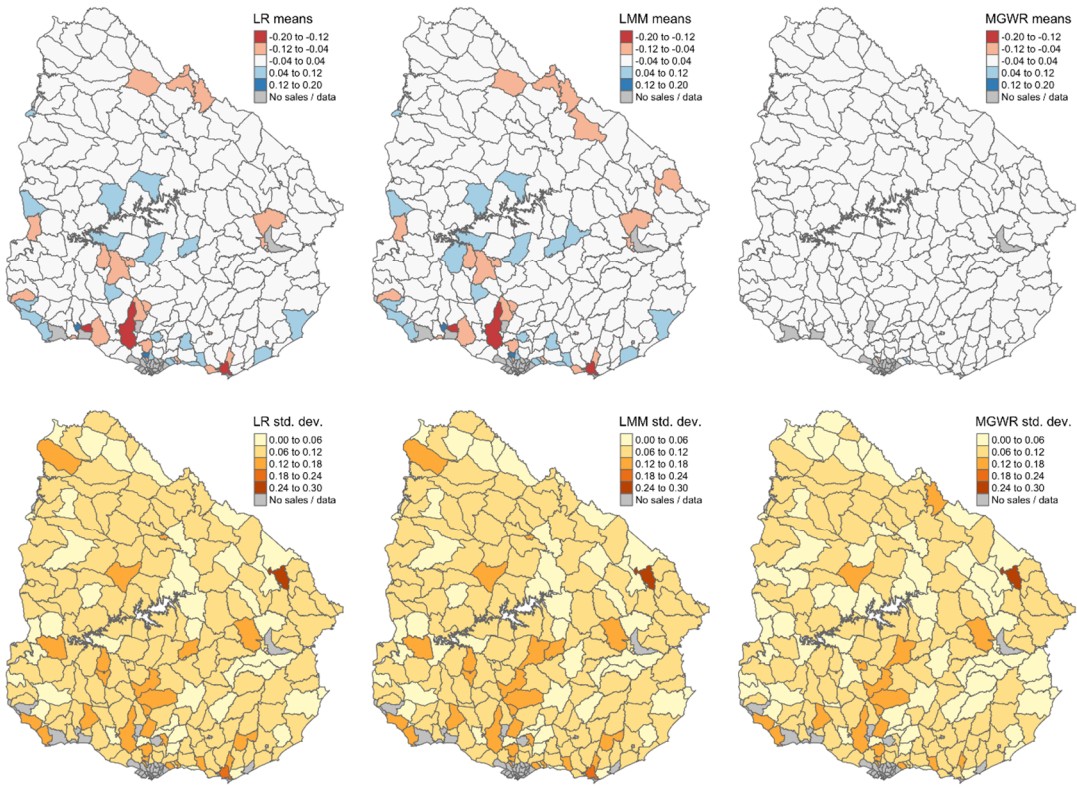

**Figure 4.** Mean and standard deviation residual maps from the three study models (LR, LMM, and MGWR). Each mean map is given with the same legend breaks from the LR fit only.

**Table 4.** Summary of study results: significant coefficients only.

| Model | Market and Cattle Characteristics * | Permanent Agro-Ecological Conditions * | Temporary Agro-Ecological Conditions * |
|---|---|---|---|
| **Coefficients Estimated as Stationary:** | | | |
| LR | PSTEER, EXRT, ORDER, ORDER2, LOTSZ, LOTSZ2, RECOM, MALE, KLW, KLW2, CLASS, BD2, BD5, CZ1 | WHC, WHC2 | T2, NDVI, NDVI2, SWR, SWR2, PAW, PAW2, SXP, NXT1, NXT2, SXT2, PXT1 |
| LMM | PSTEER, EXRT, ORDER, ORDER2, LOTSZ, LOTSZ2, RECOM, MALE, KLW, KLW2, CLASS, BD2, BD5, CZ1 | WHC, WHC2 | T2, NDVI, NDVI2, SWR2, PAW, SXP, NXT1, NXT2, SXT2, PXT1 |
| MGWR | EXRT, ORDER, ORDER2, LOTSZ, RECOM, KLW2, CLASS, BD5, CZ1 | WHC | NDVI2, SWR2, PAW, SXP, NXT1, NXT2, PXT1 |
| **Coefficients estimated as nonstationary:** | | | |
| MGWR | PSTEER, MALE, KLW, BD2 | NONE | SXT2 |

* Significant at the 5% level or lower, or at least some subset of a given SVC set significant at the 5% level or lower.

## 4. Discussion

Results from the LR hedonic model confirmed existing findings [2] for Uruguayan grass-fed cattle and the influence of market conditions and animal traits on cattle price. Results also agreed with previous findings [3] regarding the additional influence of agro-ecological factors on cattle price. These factors, reflecting soil, water, and pasture conditions, have a geographical provenance effect, influencing livestock supply and demand at different seasons of the year. The benefit of extending the latter study's hedonic LR model by estimating spatial autocorrelation effects through this study's LMM was marginal, aside from indicating a certain (spatial-) robustness to the non-spatial LR model. This was due in part to the well-specified LR model in terms of the number and breadth of predictors, where spatial autocorrelations effects commonly tend to insignificance [39]. Interestingly, the significance of surface water runoff (SWR) and available water (in quadratic form, PAW2) on cattle price found in the LR was not present in the LMM (at 5% level). This absence of significance in an LMM is common when a predictor exhibits a strong spatial trend itself [37].

Extending the same hedonic LR by estimating spatial heterogeneity effects through MGWR, uncovered new geographical insights and drivers of the Uruguayan cattle price process. Process relationships (including the intercept) were found to operate at a mixture of global (36 relationships), broadly regional (nine), broadly local (five), and highly local (one) spatial scales. Not all relationships were significant, where, of the 27 significant ones in the LMM, 17 were again estimated as global and significant in MGWR. The coefficients for SWR and PAW2 were both estimated as global in MGWR, but again were both insignificant (i.e., as in the LMM but unlike the LR, from above). This suggests that MGWR can also mitigate against spatial autocorrelation effects as an LMM does, even though MGWR is not directly designed to do so [37].

The MGWR model estimated 15 relationships as local, where four were significant everywhere (steer price (PSTEER), male (MALE), live weight (KLW), and the intercept (C)) and two were significant in some locations but not others (Angus breed (BD2) and the interaction between SWR and fall (SXT2)). The same six locally significant relationships were also significant globally for LR/LMM. Given the intercept was estimated as nonstationary (and positive and significant everywhere), it can be viewed as a pseudo autocorrelation term [37]. The remaining nine local relationships of MGWR were estimated as insignificant everywhere, one of which was significant globally for LR/LMM (pasture condition (NDVI)). By deduction, three relationships that were considered globally significant for LR/LMM were not viewed as globally or locally significant for MGWR. These predictors were the quadratics of lot size (LOTSZ2) and water holding capacity (WHC2) and fall (T2). No relationships that were estimated as globally insignificant in LR/LMM were subsequently estimated as locally (or even globally) significant in MGWR. As MGWR provided an improved fit both globally (through $R^2$ and AIC) and locally (through the residual maps of Figure 4) over LR/LMM, then much credence should be given to the change in status of many of the hedonic relationships that MGWR has uncovered. It is constructive to provide interpretations of the significant nonstationary relationships—i.e., those between cattle price and (a) PSTEER, (b) MALE, (c) KLW, (d) BD2, and (e) SXT2.

The cattle to steer price relationship was relatively weak in (the hilly) Northeast and central South regions which can be attributed to the predominance of Cow-calf and Dairy operations, respectively—where cattle are commonly finished for market elsewhere. Weak relationships were also observed in the Northwest, a region of volcanic bedrock and shallow soils, where sheep farms dominate. The cattle price to MALE relationship was relatively weak in Eastern, Northwestern and Southwestern regions. Again, the weak relationships in the East and the Northwest can be attributable to areas of poor soils (hilly in the East and shallow soils in the Northwest), where there are limitations to finishing.

Cattle price to KLW exhibited relatively strong to relatively weak relationships moving in clear trend from South to North. This can be attributed to different stocking structures for farms in the South to those

in the North, according to operation type (Cow-calf, Dairy etc.). However, the estimated SVCs change little across all areas (due to the large bandwidth of 506 km) and, therefore, little should be read into this nonstationary relationship. The relationship between cattle price and BD2 was relatively strong and significant in regions surrounding Montevideo. This can be attributed to regions that focus on Angus stock and where breeders have an established reputation for high quality beef products—entailing a premium on cattle price. The relationship of cattle price with SXT2 was relatively strong and significant in the hilly central to South regions, where pasture ground conditions tend to be good with the hilly terrain providing high water run-off.

There are caveats to this study regarding inference. First, instances of local collinearity can sometimes render local inference in GWR unreliable [40,41]. Although, in this regard, MGWR is known to be more robust to collinearity over standard GWR [37], and LR/LMM similarly did not consider collinearity issues. A penalized form of LR/LMM [42] and MGWR could have been applied, but for MGWR this would require model development, extending that given in Brunsdon et al. [43], say. Inference in MGWR may also be improved by adopting a bootstrap inferential framework [44] but would again require model development. Alternatively, a different multiscale SVC model could have been applied altogether. For example, a Bayesian model [45] or a random effects eigenvector spatial filtering model [46], but comparisons with MGWR have shown all three multiscale SVC models to have both merits and drawbacks in broadly equal quantities [46–48].

The *Mercado Común del Sur* (Mercosur, Southern Common Market) nations of Brazil, Argentina, Paraguay, and Uruguay own the largest commercial beef cattle herd in the world, estimated in 310 million heads in 2020. Together, the four Mercosur countries produce around 14 million metric tons (MMT) of beef (carcass weight) per year [49], generating 25% of world beef production. With a population of 3.45 million inhabitants and a herd of 11.5 million beef cattle heads, Uruguay has 3.3 cattle head per capita. Beef production is the most important economic activity in the country. Uruguay exports two-thirds of its total beef production. Shipments of chilled and frozen beef cuts rank in first place among Uruguay exports. With an average of 1.5 billion USD of freight on board (FOB) per year, every year, Uruguay ranks 7th to 9th in the list of largest beef exporters in the world [50,51]. Although it is a small exporter relative to the world's largest, its sanitary status is singular. All Mercosur nations are officially recognized as having negligible *Bovine Spongiform Encephalopathy* (BSE) risk by the World Organization of Animal Health (WOAH). However, Uruguay is the only one recognized with the status of Foot and Mouth Disease (FMD) free with vaccination by the same international institution [52].

Uruguay is regarded by "think tanks" as a good sample of the international beef and cattle markets due to its high dependence on beef exports and its access to most of the major import destinations in the world. Important international consultancy firms use Uruguay's beef export price index as a "proxy" of world market conditions, in their public and private reports [53]. Uruguayan beef is routinely promoted in key markets based on the health and safety benefits of natural, open-sky, grass-fed production systems [54]. Attempts to define a nationwide provenance are complicated by *observable* geographical and seasonal effects grounded in permanent and temporary agro-ecological factors that are only recognized by internal cattle markets [3]. *Unobservable* geographical effects also exist as indicated by the outputs of the spatially-explicit models of this study, especially those for MGWR.

Efforts to increase the value of Uruguay's cattle industry could either focus on improving the quality and uniformity of cattle produced in every region by reducing variability in the production of pastures all year round, or, alternatively, could acknowledge and embrace the local/regional differences indicated through MGWR, in the knowledge that local and regionally-tailored improvements could still add value to the industry, as a whole. Study results provide important geographical information to Uruguayan cattle producers about the price effects of strategic marketing and production decisions, including the relationships to breed, lot composition, cattle characteristics, cattle inspectors' recommendations, and auction prices.



As with Lanfranco and Castaño [3], this study informs on cattle price effects due to agro-ecological conditions. Permanent differences between agro-ecological regions are a function of soil productivity, water holding capacity, that, in turn, drive forage production. Each region's diversity is also fueled by seasonal variations in climate and meteorological conditions, which similarly determine yearly cycles of pasture growth (i.e., temporary differences). Uruguay's government and beef cattle supply chain have long-standing aims to increase sector-wide productivity, total exports, and price premiums for grass-fed, high-quality beef. Differentiation of Uruguayan beef in the world market based on local/regional, as done here with MGWR, rather than national provenance is one possible strategy for the sector.

## 5. Conclusions

In this study, one non-spatial and two spatial hedonic regression models of cattle prices at video auctions in Uruguay were specified for cattle produced under grass-fed systems. Predictors (50 in total) included those measuring marketing conditions, cattle characteristics, and agro-ecological factors. Results indicated that market prices were influenced, in part, by all three categories of predictors. Although many of the agro-ecological predictors are inherently spatial in nature, the incorporation of spatial effects into the estimation of the hedonic model itself, through either a spatially-autocorrelated error term (a linear mixed model) or allowing the regression coefficients to vary spatially and at different spatial scales (a multiscale geographically weighted regression) was able to provide greater geographical insight into the cattle price process than that found using the usual non-spatial model (linear regression).

Through the multiscale geographically weighted regression, which was the most informative and most accurate model, the relationships between cattle price and the predictors operated at a mixture of global, broadly regional, broadly local, and highly local spatial scales. The significance of these relationships would also vary. Of the 15 relationships estimated as nonstationary, four were significant everywhere (steer price, male, live weight, and the intercept), while two were significant in some locations but not others (Angus breed and the interaction between surface water run-off and fall). Utilizing this rich spatial information can help improve the geographical provenance of Uruguayan beef and thus maintain Uruguay's status as a key exporter of beef with respect to the health and safety benefits of natural, open-sky, grass-fed production systems. The concepts behind these results can be easily extended to almost any grassland system, worldwide.

**Supplementary Materials:** The following are available online at http://www.mdpi.com/2077-0472/10/7/299/s1, Table S1: Summaries (part 1) for the LR model, Table S2: Summaries (part 2) for the LR model, Table S3: Summaries (part 1) for the LMM, Table S4: Summaries (part 2) for the LMM, Table S5: Estimated stationary coefficients for MGWR (medians from MGWR outputs).

**Author Contributions:** Conceptualization: P.H. and B.L. (Bruno Lanfranco), and A.C.; Methodology: P.H., B.L. (Bruno Lanfranco), B.L. (Binbin Lu), and A.C.; Software: B.L. (Binbin Lu); Formal Analysis: P.H., and A.C.; Data Curation: B.L. (Bruno Lanfranco); Writing and Editing: P.H., B.L. (Bruno Lanfranco), and A.C.; Writing—Original draft: P.H. All authors have read and agreed to the published version of the manuscript.

**Funding:** This research was funded by UK BBSRC grant numbers BB/NO22408/1; BBS/E/C/000I0320; BBS/E/C/000I0330 and by UK NERC grant number NE/S009124/1. For Lanfranco, this work was also part of the normal research activities carried out at the National Agricultural Research Institute of Uruguay and financed by this institution. All analyses were undertaken using R version 3.5.0, including existing and adapted functions from the following R packages: GWmodel [55,56], scgwr [57], gstat [58], geoR [59], and nlme [31].

**Conflicts of Interest:** The authors declare no conflict of interest.

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
