# Peer review of "Influence of Geographical Effects in Hedonic Pricing Models for Grass-Fed Cattle in Uruguay"

_agriculture, doi:10.3390/agriculture10070299_

Round 1

Reviewer 1 Report

Dear author,

I have read this paper with interest and I hope you are open for some comments that may clarify some to even improve this presentation a little bit.

In the paper you have shown a very interesting research. Analysis of the factors influencing cattle prices it is necessary to have an in-depth understanding and it is very important in the perspective of forecasting the market situation.

Understanding this phenomenon can help to reduce future risks on both the supply and demand sides.

But in my opinion a few things are missing. The problem with this manuscript is several inaccuracies, which are described in review.

First of all, there is a lack of a wide overview of the literature. What the authors have proposed is primarily to refer to and quote their own works [e.g. 1, 2, 3, 20, 21, 22, 24, 30, 31, 32, 33, 34, 39, 40, 44, 46]. This is 16 references from 48 (one third).

Many concepts are only superficially referred to and should be further investigated with consistent literature references. There is no bibliographical background, however, the issues analyzed and relationships between them are discussed in the literature. Especially in relation to hedonic price analysis. The reader can't really know anything. Therefore, I believe that the manuscript must be extended to include a literature review based on many other authors.

Another thing is that the manuscript lacks an international comparison - market, prices, structure of import and export.

How does Uruguay compare to other countries? You may not need to pull in the original data, but there may be studies by others that look into that questions.

Third, I wonder if you could develop your conclusions a bit more by framing for your readers suggesting more precisely what policy or programmatic steps might be undertaken on that basis?

There is no discussion of the results obtained with the existing scientific achievements, which in my opinion this greatly reduces paper quality. This is related to the insufficient bibliographical background, references. Authors very briefly indicated to what extent their work contributed to the development of theory and practice. This is insufficient.

I think you've studied the data well but without the original data, it is difficult to discuss the results.

However, I have my doubts about the research period. Therefore, you must necessarily write an argument as to what the analysis from 2002-2009 brings to the current state of knowledge.

Especially as you write "As the cattle production process is inherently spatial, such naïve model estimations can result in both poor prediction and inference". So, have the models worked well in comparison to 2010-2020? And what is the direction in the next years? Please describe this problem.

I must add that the figures are great, they represent the problems described very well.

The title

To be considered - Influence of Geographical Effects in Hedonic Pricing Models for Grass-Fed Cattle in Uruguay

Abstract

The abstract introduces the manuscript well and briefly.

You have to indicate research period.

Introduction

You have to add an international comparison - market, prices, sales volume, structure of import and export. How does Uruguay compare to other countries? Doesn't international trade affect cattle prices in Uruguay? 

L30-31. Can they be measured by other methods?

l75 naïve -> naive

l47-48. Didn't buyers have other sources of information? Brochures, data over the Internet? Do auctions look like this nowadays? After all, 11 years have passed. The technology has developed. The past tense, if so. Please explain.

l55 “There can be significant differences” or "There were significant differences". Cited publication from 2006

Materials and Methods

l85. Lack of “Materials “

You perfectly describe the stages of research. But …

l92-93. It is unclear if prices are “Fixed” and thus adjusted for inflation. This needs to be clarified. If real prices adjusted for inflation are not included in this manuscript, then this needs to be calculated and included. The time series presented is not decades but there could have been structural changes in the cattle breeding, on the market etc. in Uruguay that may not be clear if the impacts of inflation are not accounted for in the analysis. For example, over the past 70 year time series of eggs in the U.S., nominal prices increased but real prices (adjusted for inflation) have decreased by half due to the industrialization of production, improved productivity, and lower per unit costs. Just looking at nominal prices for eggs in the U.S., one would conclude that eggs got more expensive over the past 70 years which is not true.

l96-98. How to use analyses from 2002-2009 in 2020, how to forecast? Please explain.

I would like you to add one or two more paragraphs in which you describe the indicators, why you have chosen these variables (apart from the comparison with previous articles 2, 3), why you have not chosen new, additional variables.

l110. Market and cattle variables used to modelling.

l143-144, 166-167, 240-241, 382-383 It is not allowed to be footnote in this journal. Please delete all of the footnotes or merge them into the main text.

l186 “As is commonplace” Can you give some examples?

454 (Ribeiro and Diggle, 2001) - Remove

So, the manuscript can be published after summarizing the comments and suggestions contained in this review and approval by the editor.

Reviewer 2 Report

The article is very interesting and up to date. The literature review is done carefully. I do like the idea of the research, however in my opinion the justification for the need for research in this field should be better specified.

Also the abbreviations used in the text should be explained while firstly used.

There are no limitations of the study presented.
